# Relationship between acute glucose variability and cognitive decline in type 2 diabetes: A systematic review and meta-analysis

Haiyan Chi[1,2☯], Min Song[1☯], Jinbiao Zhang[3], Junyu Zhou[1], Deshan Liu[4]*

1 Shandong University of Traditional Chinese Medicine, Jinan, Shandong, China, 2 Department of Endocrinology, Weihai Municipal Hospital, Cheeloo College of Medicine, Shandong University, Weihai, Shandong, China, 3 Department of Neurology, Weihai Municipal Hospital, Cheeloo College of Medicine, Shandong University, Weihai, Shandong, China, 4 Department of Traditional Chinese Medicine, Qilu Hospital of Shandong University, Jinan, Shandong, China

☯ These authors contributed equally to this work.
* liudeshan@sdu.edu.cn

**Data Availability Statement:** All relevant data are within the paper.

**Funding:** This work was supported by Qilu Hospital Geriatric Diseases Chinese and Western Integration

## Abstract

### Background

Cognitive decline is one of the most widespread chronic complications of diabetes, which occurs in more than half of the patients with type 2 diabetes (T2DM). Emerging evidences have suggested that glucose variability (GV) is associated with the pathogenesis of diabetic complications. However, the influence of acute GV on cognitive dysfunction in T2DM is still controversial. The aim of the study was to evaluate the association between acute GV and cognitive defect in T2DM, and provide a most recent and comprehensive summary of the evidences in this research field.

### Methods

PubMed, Cochrane library, EMBASE, Web of science, Sinomed, China National Knowledge Infrastructure (CNKI), and Wanfang were searched for articles that reported on the association between acute GV and cognitive impairment in T2DM.

### Results

9 eligible studies were included, with a total of 1263 patients with T2DM involved. Results showed that summary Fisher's $z$ value was -0.23 [95%CI (-0.39, -0.06)], suggesting statistical significance ($P = 0.006$). Summary $r$ value was -0.22 [95%CI (-0.37, -0.06)]. A lower cognitive performance was found in the subjects with greater glucose variation, which has statistical significance. Mean amplitude of glycemic excursions (MAGE) was associated with a higher risk of poor functional outcomes. Fisher's $z$ value was -0.35 [95%CI (-0.43, -0.25)], indicating statistical significance ($P = 0.011$). Sensitivity analyses by omitting individual studies showed stability of the results.

Academic School Inheritance Project (NO.2022-93-1-10).

**Competing interests:** The authors have declared that no competing interests exist.

## Conclusions

Overall, higher acute GV is associated with an increased risk of cognitive impairment in patients with T2DM. Further studies should be required to determine whether targeted intervention of reducing acute GV could prevent cognitive decline.

## Introduction

Globally, the prevalence of type 2 diabetes (T2DM) has been on the rise at an alarming rate during the past decades, due to interrelations of genetic, environmental, and other metabolic risk factors. International Diabetes Federation has estimated that there were about 41,600 new cases of diagnosed T2DM in 2021 worldwide, and there will be more than 780 million in 2045 [1–3]. As a severe and progressive disease, T2DM is strongly associated with many chronic complications, including many in the brain and nervous system.

Clinical and epidemiological studies have shown that T2DM is at an elevated risk of developing impairment of cognitive abilities, including attention, memory, concentration, executive function, information processing, visual-spatial ability, and psychomotor speed [4–6]. T2DM is characterized by constant hyperglycemia, fluctuant hyperglycemia and hypoglycemia, resulting from insulin resistance, relative insulin deficiency or both [7–9]. It seems that excessive intra-day glucose variability (GV) is closely related to a higher risk of hypoglycaemia, particularly when the average levels of HbA1c are nearly normal [10, 11].

Increasing evidences are supporting the significance of glucose variability (GV) in the progression of diabetes complications. Variability in blood glucose is the extent of oscillations that occur within a specified period. Acute or short-term GV is defined as rapid upward and downward fluctuations of glucose concentrations within or between-days [12]. Since the brain's only source of energy is glucose, it is most vulnerable to disorders of glucose metabolism.

Currently, though there is no consensus on the gold standard measurement of acute GV, indexes such as mean amplitude of glycemic excursions (MAGE), standard deviation of glucose (SD), coefficient of variation of glucose (CV), and time in range (TIR), have been mostly applied in previous studies. These parameters could be calculated from the self-detection of blood glucose or continuous glucose monitoring (CGM) [13, 14]. Whereas CGM or flash monitoring systems with glucose measurements in interstitial fluids at 5-15min time intervals is more comprehensive and widely accepted as the gold standard method for assessing acute GV [15, 16], acute GV is emerging as a possible additional index of glycemic control in recent years [17–20].

There is at present no definitive evidence supporting acute GV as an independent risk factor for cognitive impairment in T2DM. Some studies have suggested that acute GV is associated with an increased risk of cognitive decline across different domains, while others have found no association [21, 22]. On account of differences in study designs and methods to assess acute GV, it was quite difficult to draw conclusions on the consistency of the results. Furthermore, because each study has reported on specific domains using different scales, it remains unclear if acute GV has differential influences on cognitive domain. Therefore, a meta-analytic approach is especially useful for synthesizing the results of these studies. Given the high prevalence of cognitive dysfunctions in patients with T2DM and its serious consequences [23, 24], our aim is to summarize the literature and comprehensively assess the effect of acute GV on cognitive function in type 2 diabetes.

## Methods

This meta-analysis was performed by the Preferred Reporting Items for Systematic reviews and met the Meta-analysis of Observational Studies in Epidemiology (MOOSE) guidelines [25], and also referenced the Preferred Reporting Items for Systematic Reviews and Meta-analyses (PRISMA) statement [26]. The study protocol has been registered in PROSPERO (ID CRD 42023412136).

### Search strategy and selection criteria

A comprehensive literature search of all studies published until 25th December 2022 was conducted. Eligible observational studies were found by searching PubMed, Cochrane library, EMBASE, Web of science, Sinomed, CNKI and Wanfang. Resources in clinical trial register centers (http://www.clinicaltrials.gov) were also searched. The following were search terms: ("Diabetes Mellitus, Type 2" or "Diabetes Mellitus Noninsulin-Dependent" or "Diabetes Mellitus Ketosis-Resistant" or "Diabetes Mellitus Ketosis Resistant" or "Ketosis-Resistant Diabetes Mellitus" or "Diabetes Mellitus, Non Insulin Dependent" or "Diabetes Mellitus, Non-Insulin-Dependent" or "Non-Insulin-Dependent Diabetes Mellitus" or "Diabetes Mellitus, Stable" or "Stable Diabetes Mellitus" or "Diabetes Mellitus, Type II" or "NIDDM" or "Diabetes Mellitus, Noninsulin Dependent" or "Diabetes Mellitus, Maturity-Onset" or "Diabetes Mellitus, Maturity Onset" or "Maturity-Onset Diabetes Mellitus" or "Maturity Onset Diabetes Mellitus" or "MODY" or "Diabetes Mellitus, Slow-Onset" or "Diabetes Mellitus, Slow Onset" or "Slow-Onset Diabetes Mellitus" or "Type 2 Diabetes Mellitus" or "Noninsulin-Dependent Diabetes Mellitus" or "Noninsulin Dependent Diabetes Mellitus" or "Maturity-Onset Diabetes" or "Diabetes, Maturity-Onset" or "Maturity Onset Diabetes" or "Type 2 Diabetes" or "Diabetes, Type 2" or "Diabetes Mellitus Adult-Onset" or "Adult-Onset Diabetes Mellitus" or "Diabetes Mellitus, Adult Onset") And ("glucose variability" or "glycemic variability" or "glucose fluctuation" or "glucose instability"or"glycemic fluctuation") And ("Cognitive Dysfunction" or "Cognitive Dysfunctions" or "Dysfunction, Cognitive" or "Dysfunctions, Cognitive" or "Cognitive Impairments" or "Cognitive Impairment" or "Impairment, Cognitive" or "Impairments, Cognitive" or "Mild Cognitive Impairment" or "Cognitive Impairment, Mild" or "Cognitive Impairments, Mild" or "Impairment, Mild Cognitive" or "Impairments, Mild Cognitive" or "Mild Cognitive Impairments" or "Mild Neurocognitive Disorder" or "Disorder, Mild Neurocognitive" or "Disorders, Mild Neurocognitive" or "Mild Neurocognitive Disorders" or "Neurocognitive Disorder, Mild" or "Neurocognitive Disorders, Mild" or "Cognitive Decline" or "Cognitive Declines" or "Decline, Cognitive" or "Declines, Cognitive" or "Mental Deterioration" or "Deterioration, Mental" or "Deteriorations, Mental" or "Mental Deteriorations" or "Cognitions" or "Cognitive Function" or "Cognitive Functions" or "Function, Cognitive" or "Functions, Cognitive". Our PEO(Population, Exposition and Outcome) question to guide the systematic review was formulated as follows: in adult patients (18 years or above) with T2DM, study assessing the association between GV and cognitive impairment published in English or Chinese language, acute GV evaluated with one or more parameters including MAGE, SD, CV or TIR using either self-monitoring of blood glucose (SMBG) or continuous glucose monitoring (CGM). Studies were excluded if any of the following were identified: (1) insufficient information concerning GV and cognitive impairment, or the outcome cannot directly be extracted; (2) animal trials; (3) case reports, abstracts, meta-analyses, and reviews; (4) type 1 diabetes, and diabetes of special types; (5) studies conducted in pregnant women, patients who were undergoing hemodialysis, patients with critical illness or end-stage cancer.

## Data extraction and quality assessment

Two authors screened the studies and extracted data independently. All disagreements were resolved by discussion or consensus with the corresponding author. The following were included in the data: first author, publication year, country, sample size, participant characteristics, and cognitive outcomes. To increase the comprehensiveness of the inclusion process, citations, references and other related articles were sorted manually. To be specific, ten aspects deemed essential for good reporting of observational studies were assessed, including: study design and size, participants, variables, data sources, quantitative variables, confounding, subgroups and interactions, bias, missing data, and conflict of interest.

## Risk of bias assessments

The methodological quality was evaluated by the Newcastle-Ottawa Scale (NOS) [25], which consists of three parameters: Selection, Exposure, and Comparability. The score ranges from 0 to 9.

## Data synthesis

STATA 17.0 software was used for the statistical analysis. Cochrane's Q-test was performed and the $I^2$ statistic was estimated as previously described to evaluate the extent of between-study heterogeneity [26, 27]. A fixed effect model was adopted for data synthesis, if $I^2 < 50\%$, and a random-effect model was adopted, if $I^2 \geq 50\%$ [27]. The association between GV and the risk of cognitive impairment was presented as correlation coefficients $r$. Pearson's $r$ and non-Pearson's $r$, such as Spearman's Rho, which can subsequently be transformed into Pearson correlation values, were all included. Pooled estimates were calculated by transforming each $r$ into Fisher's $z$ values. The resulting values were then weighted with the inverse of the variance of the correlation coefficients [28]. The 95% confidence intervals(CIs) of the pooled weighted Fisher's $z$ values were calculated, after which all the values were back-transformed to $r$ using the following formula [29, 30].

$$\text{Fisher's } z = 0.5 \times \ln\frac{1+r}{1-r} \qquad \text{①}$$

$$Vz = 1 / (n - 3) \qquad \text{②}$$

$$S_E = \sqrt{Vz} \qquad \text{③}$$

$$\text{Summary } r = (e^{2Z} - 1) / (e^{2Z} + 1) \qquad \text{④}$$

Correlation was evaluated comprehensively with the range of absolute value of summary $r$ as follows: $0.1 \sim 0.3$, $0.3 \sim 0.5$, $0.5 \sim 1.0$ indicates small, moderate and large correlation, respectively [31]. Subgroup analyses were conducted based on the following prespecified factors: MAGE, SD and CV. Forest plots were used to characterize the results of various studies, and sensitivity analysis was conducted to explore the stability and reliability of the analytical results. Egger's test and Begg's regression test [32, 33] were both performed to assume potential publication bias, and a $P$-value of $< 0.05$ suggested statistical significance.

## Results

### Results of the literature search

A total of 307 articles were retrieved by searching electronic databases while removing duplication. 261 were further excluded due to lack of relevance. The remaining 46 studies were screened with full texts, and 37 were further removed for the reasons in Fig 1. For every study included, the retrieved data were as follows: first author's name, publication year, country, samples, age, gender, GV index and duration, cognitive outcomes, and adjustments for potential confounding factors. Finally, 9 studies [34–42] were available for meta-analysis. The literature review process is shown in Fig 1.

### Characteristics of the included studies

As shown in Table 1, 9 observational studies including 1263 patients with T2DM were included. The studies were performed in Japan, South Korea, China and Italy, and published between 2010 and 2022. The proportions of men ranged between 32.4% and 77.8%. Acute GV was evaluated with MAGE, SD, CV, and TIR. The duration of acute GV measures varied between 48 hours and 14 days, except for two studies. Four articles used MMSE [43], and four used MoCA [44], and one used both the MMSE and MoCA for global neuropsychological assessments. Variables such as age, sex, education and HbA1c etc. were adjusted to different degrees among the included studies. The NOS scores of the included studies were seven to nine, suggesting good quality, as showed in S1 File.

### Meta-analysis results

13 acute GV indexes including MAGE, SD and CV were extracted from nine pieces of literature. The heterogeneity test showed that there was significant heterogeneity among all studies ($I^2$ = 91.7%, $P<0.01$), so the random effects model was used. Results showed that summary Fisher's $z$ value was -0.23[95%CI (-0.39, -0.06)], suggesting statistical significance ($P$ = 0.006). Further calculation results demonstrated that, a lower cognitive performance was found in the

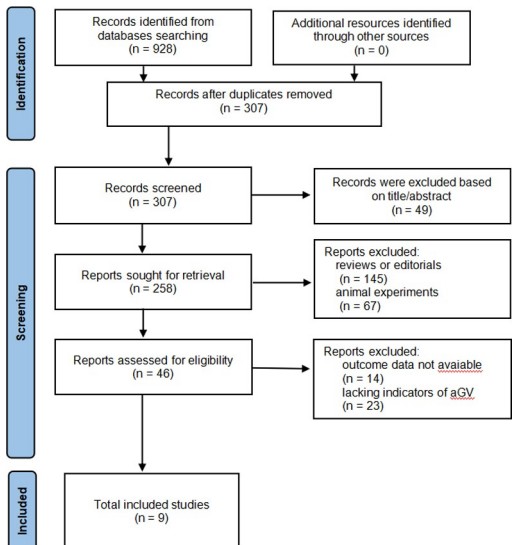

**Fig 1. Flowchart of database search and study identification.**

**Table 1. Characteristics of the included studies in the meta-analysis of the association between acute GV and cognitive impairment.**

| Author, year | Country | Sample size | Mean age (years) | Male (%) | GV index | Duration of GV measurements | Cognitive outcomes | Adjusted Variables | NOS |
|---|---|---|---|---|---|---|---|---|---|
| Taiki Sugimoto, 2022 [34] | Japan | 100 | 77.0±4.2 | 61.0 | TIR, CV | 14 days | MoCA, TMT, DST | age, sex, education, BMI, BP, hemorrheologic, diabetes duration, HbA1c, use of antidiabetes drugs | 8 |
| Wenqing Xia, 2020 [35] | China | 97 | 56.0±7.2 | 55.5 | MAGE | 3 days | MMSE, MoCA, AVLT, CFT, DST, TMT, CDT | age, sex, education, BP, hemorrheologic, HbA1c | 8 |
| Xianghong Xie, 2020 [40] | China | 80 | 69.4±6.5 | 61.2 | MAGE | 72 hours | MoCA | age, sex, BMI, BP, HbA1c | 9 |
| Jae-Sung Lim, 2018 [36] | South Korea | 388 | 63.0±12.1 | 64.4 | MAGE, SD | NA | MMSE, K-VCIHS-NP | age, sex, education, BP, hemorrheologic, use of antidiabetes drugs | 7 |
| Jian Yi, 2018 [42] | China | 81 | 60.5±7.6 | 55.6 | MAGE, SD | 2 days | MoCA | age, sex, education, diabetes duration | 7 |
| Chulho Kim, 2015 [39] | South Korea | 68 | 70.9±5.9 | 32.4 | CV | NA | MMSE, VLT, DST, CDR | age, years in fulltime education, other demographic factors, and vascular risk factors | 7 |
| ZHONG Yuan, 2012 [37] | China | 248 | 80.2±9.6 | 77.8 | MAGE | 3 days | MMSE, CDR, GDS | age, sex, education, BP, HbA1c, diabetes duration | 8 |
| Mengmeng Li, 2022 [41] | China | 80 | 68.6±6.0 | 53.6 | MAGE, SD | 3 days | MoCA | HbA1c, BMI | 7 |
| Maria Rosaria Rizzo, 2010 [38] | Italy | 121 | 78.0±6.7 | 37.7 | MAGE, SD | 48 hours | MMSE, TMT | age, sex, BMI, BP, diabetes duration | 7 |

NOS Newcastle-Ottawa Scale; MoCA Montreal Cognitive Assessment; MMSE Mini-Mental State Examination; NA not applicable; TMT trail-making Test; DST Digit Span test; AVLT Auditory Verbal Learning Test; CDT Clock Drawing Test; K-VCIHS-NP Korean version of the Vascular Cognitive Impairment Harmonization Standards-Neuropsychological Protocol; GDS Geriatric Depression Scale; CDR Clinical dementia rating; GDS Global deterioration scale; BMI body mass index; BP blood pressure; WHR Waist-to-Hip Ratio.

subjects with greater glucose variation, which has statistical significance, and the summary $r$ value was -0.22[95%CI (-0.37, -0.06)]. Pooled subgroup results showed that higher acute GV measured by MAGE was associated with a higher risk of poor functional outcome, and summary Fisher's $z$ value was -0.35[95%CI (-0.43, -0.25)], indicating statistical significance ($P$ = 0.011, as shown in Fig 2). Four of the articles included in this study used the MMSE, four used the MoCA, and one used both. Subgroup analysis showed a summary r value of-0.22 [95%CI (-0.37, -0.06)] using the MMSE and a summary r value of-0.31 [95%CI (-0.46, -0.07)] using the MoCA, and the difference in summary effect sizes using different scales had no statistical significance. Further sensitivity analyses by omitting individual studies showed consistent results (as shown in Fig 3).

## Publication bias

Publication bias for MAGE was assessed by Egger's test and Begg's test and no statistical publication bias was detected ($P$ = 0.089, $P$ = 0.296). Publication biases for the meta-analyses with SD and CV were difficult to estimate because only two to four studies were included.

## Discussion

To the best of my knowledge, this is the first systematic review on this topic. As the gold-standard measurement in glycaemic control, HbA1c has been a vital marker of glycaemic exposure over the past 2–3 months. However, it cannot shed light on the inter- or intra-day fluctuation of glucose and often lags behind the changes in glucose. Patients with identical levels of HbA1c

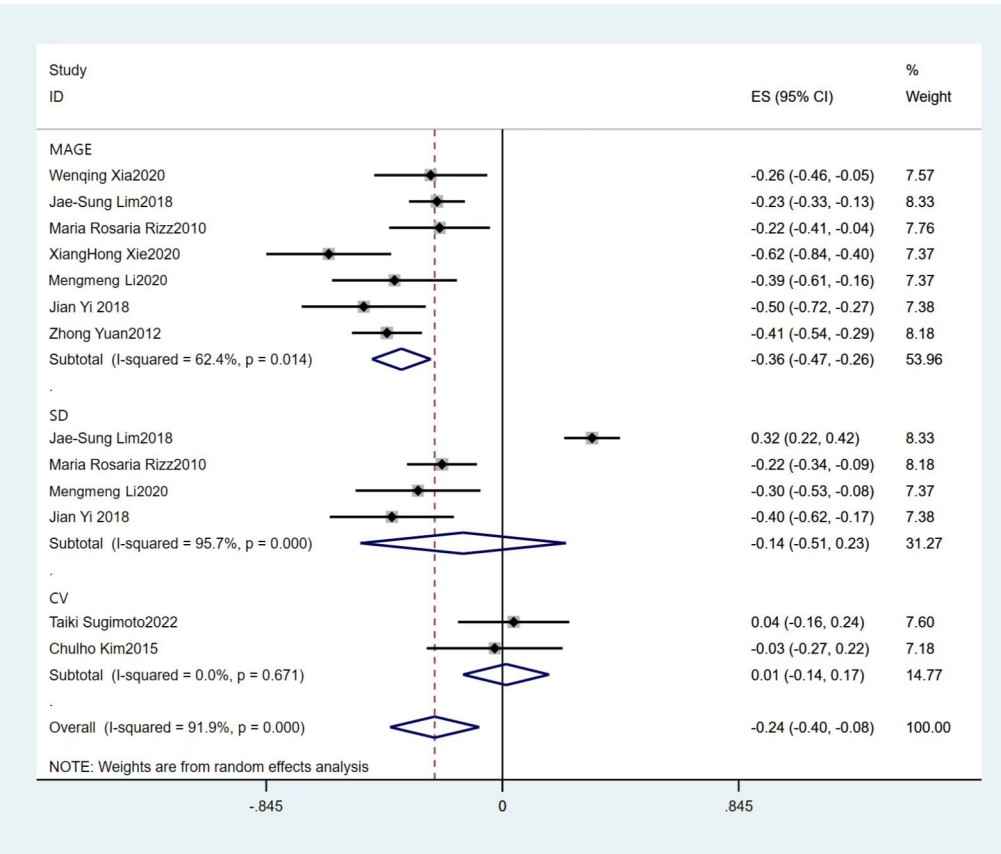

**Fig 2. Forest plots of meta-analysis of the correlation between acute GV and cognitive dysfunction.**

can differ widely in the range of glucose profiles, as well as in the rates of micro- and macrovascular complications [45, 46].

Acute GV may be an accelerator for the deleterious effects of persistent ambient hyperglycaemia. Despite its clinical importance, the correlation between acute glucose variability and cognitive impairment in T2DM has not been much explored in clinical practice. Only 9 papers in accordance with the inclusion and exclusion criteria were enrolled in the meta-analysis. The results showed that indicators of acute blood glucose variation, such as MAGE, SD, and CV, were closely associated with cognitive decline in patients with type 2 diabetes. Traditional indexes of acute GV include MAGE, SD, CV, absolute means of daily difference(MODD), largest amplitude of glycemic excursions (LAGE), etc, among which MAGE is the most commonly used term.

MAGE is an arithmetic average obtained by calculating the blood glucose fluctuations within 24 hour period when removing small fluctuations that do not exceed the threshold (normally 1SD); the direction of the calculation is determined by the first countable excursion. The SD is the standard deviation of all the glucose values during the monitoring period to evaluate the overall deviation from the mean glucose levels. CV is SD divided by the mean glucose and multiplying by 100 to get a percentage, evaluating the overall deviation from the average glucose value, and is recognized internationally as a more direct way demonstrating the fluctuation of blood glucose. The above indicators reflect acute GV from different facets.

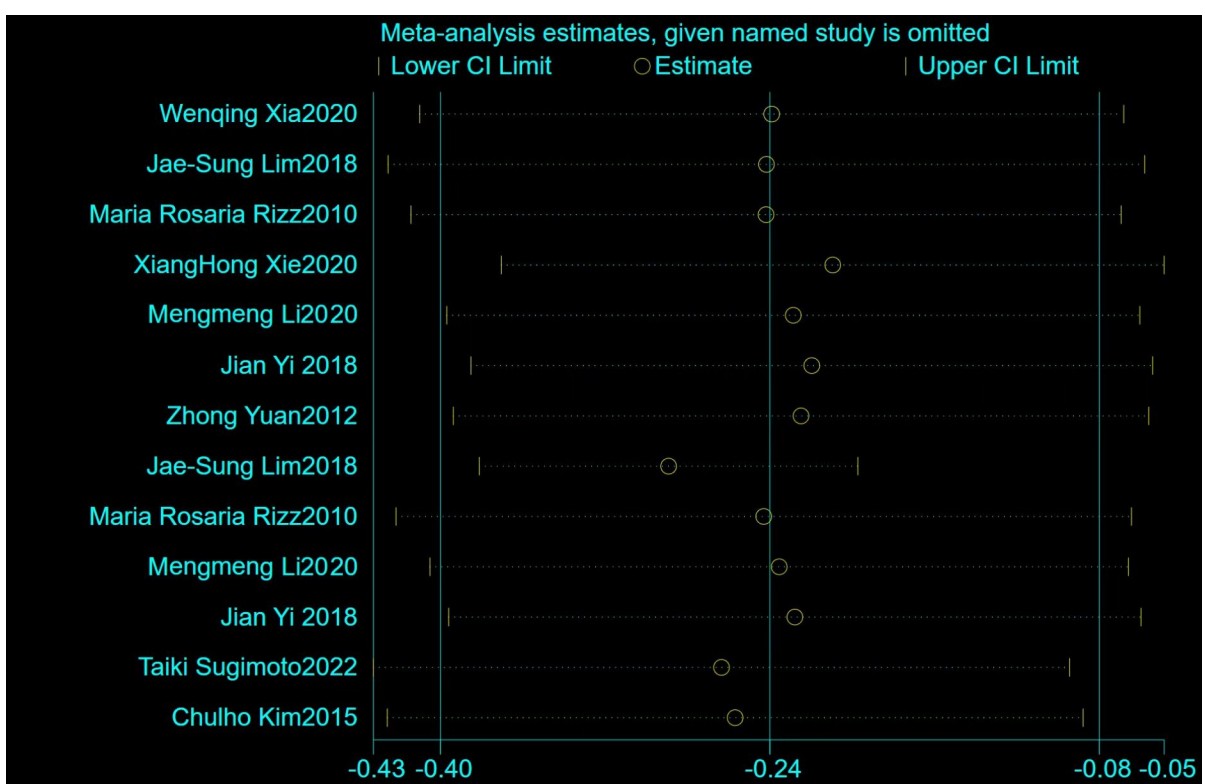

**Fig 3. Results of sensitivity analyses.**

With improvement in accuracy and sophistication, flash glucose monitor (FGM) has been widely used in clinical practice. The relationship between such core indicators in the FGM map, as TIR (time in range), and chronic complications of diabetes has attracted increasing attention. American Diabetes Association(ADA) included TIR as an indicator to evaluate glucose fluctuations for the first time in 2019, with the recommended range≥70% [47, 48]. It has been found that excessive intra-day glucose fluctuations were associated with a higher risk of hypoglycemia, particularly when the average HbA1c levels were near normal [49–51]. The downward and upward fluctuations around the mean daily glucose value contributed similarly to the short-term GV.

With regard to assessing acute GV, different indexes were adopted in different studies. MAGE was used in three articles [35, 37, 40], MAGE and SD were used in four articles [36, 38, 41, 42], CV was used in one article [39], and TIR and CV were used in one article [34]. On such basis, an exploratory subgroup analysis was performed. Results showed that although the association between higher MAGE and increased risk of cognitive dysfunction was not significantly affected by study characteristics including design, sample size, age of the patients, gender, duration for GV measuring, or study quality, the association was mainly observed in studies with older patients. Based on a random-effects model in this study, the results showed that higher acute GV was associated with an increased risk of cognitive decline in T2DM. Research quality assessments showed that the quality of the reports was generally good, scoring 7–9 by NOS.

Results of subgroup analysis for the association between MAGE and the risk of cognitive dysfunction were basically consistent. The robustness of the study was evidenced by the consistent results of meta-analyses using different parameters for acute GV, including SD and CV.

However, because there was only one article probing into the relationship between TIR and cognitive dysfunction, so subgroup analysis did not proceed. A closer inspection of this finding revealed that the meta-analytic results for the subset of studies with SD and CV were only based on two or four studies, respectively. Therefore, further explorations of relevant studies are necessary to elucidate and support this finding.

The duration of GV measurements was not mentioned in two articles [36, 39], and in other seven articles, it was described as 48 hours or 2 days in two articles [38, 42], 72 hours or 3 days in four [35, 37, 40, 41], and 14 days in one [34]. Generally speaking, the longer the time, the higher the accuracy of indicators. However, due to the limited literature included, further subgroup analysis of the evaluation time of GV was not performed.

In the literature included in this study, 4 used the MMSE, 4 used the MoCA, and 1 used both the MMSE and MoCA. The MMSE and MoCA are the most commonly used tests to assess cognitive function as a whole. Both tests use a 30-point scale. Of the two, the MMSE is comparatively simple, and more suitable for rapid detection of cognitive impairment. It demonstrates good sensitivity and specificity in detecting severe cognitive impairment, but presents a low detection rate for mild cognitive impairment. While the MoCA, with item units and scoring criteria modified based on the MMSE, is more difficult and time-consuming, but has a high sensitivity to the screening and diagnosis of mild cognitive impairment. In clinical practice, MMSE and MoCA are often jointly administered, and comprehensive evaluation should be carried out based on the subject's medical history. Other screening tests, such as CDT, TMT, DST, AVLT, CFT, etc., are designed to assess certain aspects of cognitive function and are generally used as supplements.

Taken together, these results indicated that higher acute GV may be a predictor of cognitive dysfunction in patients with T2DM. Although mean blood glucose levels and HbA1c are typically used to assess the status of T2DM, the avoidance of acute GV may be a key strategy to prevent further progression of cognitive impairment in patients with T2DM.

Both hyperglycemia and hypoglycemia can lead to cognitive decline and even dementia. Hyperglycemia accelerates the production of advanced glycation end products in blood and tissues [52], resulting in oxidative stress and inflammation amplification. Hyperglycemia will increase the production of free radicals and circulating cytokines, damage antioxidant and innate immune defense, activate NF-κB pathway and further aggravate inflammation and oxidative stress. With increased accumulation of amyloid precursor protein and amyloid beta plaque formation in neuron cells, the effects on brain function is detrimental [53].

Short-term mild hypoglycemia can lead to reversible cognitive impairment, while persistent or severe hypoglycemia can disrupt the brain's energy metabolism, leading to hippocampal atrophy and permanent neuronal damage, inducing microglial cell activation, further damaging brain structure and leading to changes in cognitive function [54]. Under hypoglycemic stress, the levels of pro-inflammatory factors (such as TNF-α, IL-1β and IL-6) are up-regulated, leading to the occurrence of acute inflammatory state [55]. In the meantime, it increases reactive oxygen species in hippocampal mitochondria of DM rats exposed to recurrent hypoglycemia, which aggravates blood-brain barrier damage, brain edema and pericellular damage in DM mice [56–58].

Several possible mechanisms may contribute to cognitive impairment associated with GV in T2DM. For both humans and animals, the hippocampus plays an important role in learning and memory. It has been confirmed by a large number of clinical and fundamental experiments that various morphological, and histological changes could be found in the hippocampi of diabetic patients with cognitive and memory decline [59–61]. By setting up a model of postprandial GV in female GK rats, it was found that acute GV could impair spatial orientation

and memory with increased levels of TNF-α and IL-1β and neuronal apoptosis in hippocampal tissue by Bcl-2/Bax/c-myc signaling [62].

Pathophysiologically, glucose oscillation has been associated with increased oxidative stress, endothelial dysfunction, and inflammatory response, three key factors that have been involved in the development of cognitive dysfunction through a variety of downstream mechanisms, including detrimental connectivity of the brain network secondary to the dysfunction of Tau phosphorylation, plaque formation, synaptic transmission modulation, and dystrophic neurite growth [63–65]. The underlying reasons behind this finding have yet to be elucidated. In particular, GV is associated with increased production of reactive oxygen species, which in turn, causes glucose-mediated vascular damage in the central nervous system [66–69].

Acute GV, which reflects the fluctuation in glycemia within or between days, has been proposed as a strong risk factor for poor outcome in patients with T2DM. In multivariate analysis, the ratio of glycoalbumin/hemoglobin (an indicator of blood glucose fluctuations) was independently associated with white matter lesions, indicating that large fluctuations in blood glucose levels may lead to white matter lesions, cognitive dysfunction, and decline in instrumental activities of daily living [70].

Nevertheless, the influence of glucose variability on the risk of diabetes-associated complications remains controversial. The changes in cognitive function caused by interval sprinting were not associated with changes in blood glucose level in 26 males who are regularly engaged in recreational activities [21]. Merve Uyar et al. found that the level of HbA1c, which represented the degree of long-term control of blood sugar, was the most important influencing factor for diabetes-related axon damage [22]. A randomized double-blind controlled study of 24 volunteers suggested that the accuracy of word memory and spatial memory tasks could improve by the oral administration of 25–60 g of glucose, compared with the placebo group [71].

Through the rat models, it was also found that glucose control itself played a stronger role than glucose fluctuation in terms of preventing peripheral nerve injury, and that intra-day glucose fluctuation had little effect on the progression of peripheral neuropathy in diabetic rats [72]. A PRANDIAL strategy has also demonstrated that lower intraday GV does not result in a reduction in cardiovascular outcomes, and the result could not support the hypothesis that targeting GV would be beneficial in reducing subsequent cardiovascular events [73].

Therefore, it could be hypothesized that the influence of increased acute GV may be multifaceted, comprehensive and complex. Our study observed that the acute GV was closely related to the cognitive function of type 2 diabetes patients. Therefore, in order to reduce the decline of cognitive function of type 2 diabetes patients and improve their quality of life, clinicians need to, besides paying attention to reaching the overall blood glucose standard, make efforts to reduce the happening of acute GV when formulating the hypoglycemic plan. Future studies are warranted to determine whether reducing acute GV could improve cognitive function in T2DM.

## Limitations

There are some limitations to the current study. Firstly, owing to the small number of available studies on this topic, it was impossible to stratify based on ethnic composition. Secondly, most studies have used MAGE as the parameter for acute GV, and studies of other parameters, such as SD and CV, are limited. TIR, as a new measure of blood glucose fluctuation, was only used in one study, so the Meta analysis between TIR and the risk of cognitive impairment was not included in the study. Therefore, the results of meta-analyses with parameters of SD, CV and TIR should be validated in the future. Finally, the cross-sectional design was unsuitable for addressing a cause–effect relationship between acute GV and cognitive dysfunction in T2DM, hence further prospective studies are required to address this issue.

## Conclusion

To sum up, the cognitive function of patients with T2DM is closely related to acute GV. Acute GV may serve as a risk factor or an accelerator for the development and progression of cognitive impairment. More prospective studies should be considered to determine whether targeted intervention to reduce acute GV could prevent cognitive decline in these patients. Clinicians should pay attention to reducing blood glucose fluctuations and hypoglycemic as well as hyperglycemic events.

## Supporting information

**S1 Checklist. PRISMA 2020 checklist.**
(DOCX)

**S1 File. Newcastle - Ottawa quality assessment scale case control studies.**
(DOC)

## Acknowledgments

We thank Shandong University of Traditional Chinese Medicine for providing the infrastructure and facilities.

## Author Contributions

**Conceptualization:** Deshan Liu.

**Data curation:** Haiyan Chi.

**Formal analysis:** Min Song.

**Methodology:** Jinbiao Zhang.

**Software:** Haiyan Chi.

**Visualization:** Junyu Zhou.

**Writing – original draft:** Min Song.

**Writing – review & editing:** Deshan Liu.

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
