## [Decision Letter · Decision Letter 0]

16 May 2023

PONE-D-23-09111Correlation between Acute Glucose Variability and Cognitive decline in Type 2 Diabetes: A Systematic Review and Meta-AnalysisPLOS ONE

Dear Dr. Chi,

Thank you for submitting your manuscript to PLOS ONE. After careful consideration, we feel that it has merit but does not fully meet PLOS ONE’s publication criteria as it currently stands. Therefore, we invite you to submit a revised version of the manuscript that addresses the points raised during the review process.

We look forward to receiving your revised manuscript.

Kind regards,

Victor Manuel Mendoza-Nuñez, PhD

Academic Editor

PLOS ONE

Journal Requirements:

When submitting your revision, we need you to address these additional requirements.2.

- 10.1055/a-1837-0141

- https://doi.org/10.1186/s13098-022-00826-9

- 10.1001/jamaneurol.2017.2180

In your revision ensure you cite all your sources (including your own works), and quote or rephrase any duplicated text outside the methods section. Further consideration is dependent on these concerns being addressed.

“This work was supported by Qilu Hospital Geriatric Diseases Chinese and Western Integration Academic School Inheritance Project(NO.2022-93-1-10) and The Development Plan of Medical Sciences of Shandong Province (2019 WS225).”

Reviewers' comments:

Reviewer's Responses to Questions

**Comments to the Author**

1. Is the manuscript technically sound, and do the data support the conclusions?

Reviewer #1: Partly

Reviewer #2: Yes

2. Has the statistical analysis been performed appropriately and rigorously? 

Reviewer #1: Yes

Reviewer #2: Yes

3. Have the authors made all data underlying the findings in their manuscript fully available?

Reviewer #1: Yes

Reviewer #2: Yes

4. Is the manuscript presented in an intelligible fashion and written in standard English?

Reviewer #1: Yes

Reviewer #2: Yes

5. Review Comments to the Author

Reviewer #1: The study identified the relationship between acute glucose variability and cognitive decline in people with type 2 diabetes. The introduction, method and discussion sections meet the information requirements for publication. In the results section, it is convenient to make some modifications to demonstrate the consistency of what the authors found, and to making modifications in the wording and figures to highlight the findings. Considering the above, major and minor modifications are suggested.

1. Major modifications

1.1. In the results section, few papers are included. The document indicated that they found "928" articles and after removing duplicates, 307. Perform a search in PubMed using the criteria proposed by the authors, including the filter by date, I found 2305 articles. If a search strategy was used in each database, it is necessary to place it. On the other hand, the number of duplicates that are reported seems high to me, it would be useful to indicate in figure 1 how many articles were found in each database.

1.2. In the results section, it failed to mention the cognitive instruments used to identify the effect of glucose variability in a meta-analysis. Table 1 indicates screening instruments such as the MMSE and the MoCA, as well as instruments to assess more specific cognitive functions such as the Trail Making Test. When carrying out the meta-analysis, they do not indicate whether they included all the indicators or only some. In addition, they could carry out the analysis by subgroups of the cognitive indicators. In case of not finding differences between cognitive instruments, just mention it in results and in discussion analyze if an analysis by specific cognitive processes or with a general instrument is different.

1.3. Evaluate the advisability of changing "cognitive imparitment" and "risk of cognitive impairment" to "cognitive decline" in the title and the different sections. The term "cognitive impairment" can be confused with the diagnosis of "mild cognitive impairment" which has specific characteristics that are not included in the study performed.

1.4. It would be interesting to highlight the conditions in which the relationship between glycemic variability and cognitive performance is found, particularly when comparing the conditions of hyperglycemia and hypoglycemia. If it is in both, make an explanation of the mechanisms under discussion separately.

1.5. In the title, change "correlation" to "relationship". The word correlation is more associated with a statistical term.

2. Minor modifications

2.1. Abstract

2.1.1. In results, you need to write what is related, not just the statistical value found. Example: "a lower cognitive performance was found in the subjects with greater glucose variation, which has statistical significance..."

2.2 Introduction

2.2.1. To facilitate reading, write paragraphs in a size of 4 to 8 lines.

2.2.2. In the last paragraph it is indicated that there are inconsistencies in the studies of the effect of glucose variability on cognitive variability, however, it is necessary to place references that support these affirmations. In this section you can consider other related topics, for example, cardiovascular health. This allows to write the inconsistencies avoiding including the articles of the results section.

2.2.3 End the introduction with the purpose of the article. Information on social relevance (in which he writes “Given the high prevalence of cognitive dysfunctions in patients with T2DM and its serious consequences [24, 25], it is crucial to explore the role of acute GV as a modifiable risk factor”) can match it with the information presented at the beginning of the document.

2.2.4. The following information can be removed without affecting the text content:

“Central nervous system-related complications of diabetes have been known for more than 100 years by clinicians and researchers as patients have constantly complained of memory loss and inattentiveness. With a growing epidemic of diabetes and an aging population, neural complications of T2DM are expected to rise rapidly and become challenges for future public health implications.”

2.2.5 In the introduction, the abbreviation "GV" is used before indicating what it means.

2.3 Method

2.3.1. In the "Search strategy and selection criteria" section, include the question with the PEO (Population, Exposition and Outcome) format.

2.3.2. In the "bias risk assessment" section, include the score range of the Newcastle-Ottawa Scale (0-8), to facilitate the reader's interpretation. Risk of bias assessment table can be added as supplementary material.

2.3.3 In the results synthesis section, reference values are indicated to interpret "r" as correlation, it is pertinent to place reference values as effect size, in which r= 0.1, r=0.3 and r=0.5 are considered small, moderate and large respectively.

2.4 Results

2.3.1. Verify data. In the text it indicates that 64 studies were reviewed, but in figure 1 46 reviewed studies are mentioned.

2.3.2. Mention the relationship of the data found and the impact of the association, before the significance values, for example, "a lower cognitive performance was found in the subjects with greater glucose variation, which has statistical significance... and an effect size…”

2.3.3. Also mention the negative results of the glucose variability indicators in the text.

2.3.4. Figure 2 can be omitted.

2.3.5. Add a subgroup analysis with the MMSE and MoCA..

2.4 Discussion

2.4.1. Write paragraphs of 4 to 8 lines to facilitate reading

2.4.2. It is necessary to write in the first paragraph the main results, that is, the relationship between glycemic variability and cognitive performance. Not only that 9 studies were found.

2.4.3 It is not relevant to indicate the brand of the glucose monitor (“Abbott” flash glucose monitor) and it creates a conflict of interest, so I suggest removing it.

2.4.4. Differences were found between the indicators of glucose variability and its relationship with cognitive functions. Although there are few studies, a description or explanation of why there are differences between the indicators can be made.

2.4.5. The paragraph on the controversies regarding the relationship of glucose variability to complications delves into cardiac problems that are not relevant to the topic, so it is necessary to explicitly state their relevance to cognitive performance or eliminate that information.

2.4.5. The writing can be finished highlighting the main results and adding the clinical implications.

Reviewer #2: The manuscript is technically sound, its methodology is appropriate, the statistical analysis was adequate, all the results obtained are presented, and it meets the publisher's criteria for publication.

6. PLOS authors have the option to publish the peer review history of their article (what does this mean?). If published, this will include your full peer review and any attached files.

Reviewer #1: **Yes: **Sánchez-Nieto José Miguel

Reviewer #2: No

---

## [Author Response · Author response to Decision Letter 0]

17 Jul 2023

Dear Editor, 

 Thank you very much for your comments and suggestions.

We have revised the manuscript, according to the comments of reviewers and editor, and responded, point by point, to the comments as list below. I have marked all the amendments in the manuscript with red color.

 I would like to re-submit this revised manuscript to the Journal of PLOS ONE, and hope it is acceptable for publication in the journal.

 Looking forward to hearing from you soon.

 Yours sincerely,

Liu, Deshan.

Department of Traditional Chinese Medicine, Qilu Hospital of Shandong University, Jinan, Shandong 250012, PR China

E-mail address: liudeshan@sdu.edu.cn.

1. Major modifications

1.1. In the results section, few papers are included. The document indicated that they found "928" articles and after removing duplicates, 307. Perform a search in PubMed using the criteria proposed by the authors, including the filter by date, I found 2305 articles. If a search strategy was used in each database, it is necessary to place it. On the other hand, the number of duplicates that are reported seems high to me, it would be useful to indicate in figure 1 how many articles were found in each database.

Response: Only parts of the search strategies are shown in the original text, and we now add all the search strategies to the main body of the text.

1.2. In the results section, it failed to mention the cognitive instruments used to identify the effect of glucose variability in a meta-analysis. Table 1 indicates screening instruments such as the MMSE and the MoCA, as well as instruments to assess more specific cognitive functions such as the Trail Making Test. When carrying out the meta-analysis, they do not indicate whether they included all the indicators or only some. In addition, they could carry out the analysis by subgroups of the cognitive indicators. In case of not finding differences between cognitive instruments, just mention it in results and in discussion analyze if an analysis by specific cognitive processes or with a general instrument is different.

Response: The MMSE and MoCA are the most commonly used tests to assess cognitive function as a whole. Both tests use a 30-point scale.. Of the two, the MMSE is comparatively simple, and more suitable for rapid detection of cognitive impairment. It demonstrates good sensitivity and specificity in detecting severe cognitive impairment, but presents a low detection rate for mild cognitive impairment. While the the MoCA, with item units and scoring criteria modified based on the MMSE, is more difficult and time-consuming, but has a high sensitivity to the screening and diagnosis of mild cognitive impairment. In clinical practice, MMSE and MoCA are often jointly administered, and comprehensive evaluation should be carried out based on the subject's medical history. Other screening tests, such as CDT, TMT, DST, AVLT, CFT, etc., are designed to assess certain aspects of cognitive function and are generally used as supplements. In the literature included in this study, 4 used the MMSE, 4 used the MoCA, and 1 used both the MMSE and MoCA. A discussion on the usage of tests has been added in the discussion section.

1.3. Evaluate the advisability of changing "cognitive imparitment" and "risk of cognitive impairment" to "cognitive decline" in the title and the different sections. The term "cognitive impairment" can be confused with the diagnosis of "mild cognitive impairment" which has specific characteristics that are not included in the study performed.

Response: "cognitive impairment" and "cognitive decline" are similar in meanings. All the included literature used scales for the overall assessment of cognitive function, such as MMSE and MoCA. Both are based on a total score of 30 points. In the MMSE, 27 to 30 suggests normal cognition, 21-26 indicates mild cognitive dysfunction, 10-20 is moderate, and 9 points or lower is severe cognitive dysfunction. In the MoCA, the scores are grouped as 26 to 30, 18 to 25, 10 to 17and 9 points or lower. The literature considers a score of 27 in MMSE or 26 in MoCA as the criteria for cognitive decline.

1.4. It would be interesting to highlight the conditions in which the relationship between glycemic variability and cognitive performance is found, particularly when comparing the conditions of hyperglycemia and hypoglycemia. If it is in both, make an explanation of the mechanisms under discussion separately.

Response: Both hyperglycemia and hypoglycemia have an impact on cognitive function, we have added the effects of hyperglycemia and hypoglycemia on cognitive function and the possible mechanisms in the discussion section. 

1.5. In the title, change "correlation" to "relationship". The word correlation is more associated with a statistical term.

Response: The title of the paper is revised to: Relationship between Acute Glucose Variability and Cognitive decline in Type 2 Diabetes: A Systematic Review and Meta-Analysis.

2. Minor modifications

2.1. Abstract

2.1.1. In results, you need to write what is related, not just the statistical value found. Example: "a lower cognitive performance was found in the subjects with greater glucose variation, which has statistical significance..."

Response: On the basis of statistical values, we have added the corresponding content.

2.2. Introduction

2.2.1. To facilitate reading, write paragraphs in a size of 4 to 8 lines.

Response: For readers’ convenience, the content of the Introduction has been refined and reduced to 4 to 8 lines.

2.2.2. In the last paragraph it is indicated that there are inconsistencies in the studies of the effect of glucose variability on cognitive variability, however, it is necessary to place references that support these affirmations. In this section you can consider other related topics, for example, cardiovascular health. This allows to write the inconsistencies avoiding including the articles of the results section.

Response: In the last paragraph it is indicated that there are inconsistencies in the studies of the effect of glucose variability on cognitive variability, we have added the relevant references.

2.2.3. End the introduction with the purpose of the article. Information on social relevance (in which he writes “Given the high prevalence of cognitive dysfunctions in patients with T2DM and its serious consequences [24, 25], it is crucial to explore the role of acute GV as a modifiable risk factor”) can match it with the information presented at the beginning of the document.

Response: The end of the introduction has been changed to: our aim is to make a summary of the literature and comprehensively assess the effect of acute GV on cognitive function in type 2 diabetes.

2.2.4. The following information can be removed without affecting the text content:

“Central nervous system-related complications of diabetes have been known for more than 100 years by clinicians and researchers as patients have constantly complained of memory loss and inattentiveness. With a growing epidemic of diabetes and an aging population, neural complications of T2DM are expected to rise rapidly and become challenges for future public health implications.”

Response: The above content has been deleted.

2.2.5. In the introduction, the abbreviation "GV" is used before indicating what it means.

Response: In the introduction, we have added the explanation of the abbreviation GV.

2.3. Method

2.3.1. In the "Search strategy and selection criteria" section, include the question with the PEO (Population, Exposition and Outcome) format.

Response: We have added the format of PEO (Population, Exposure and Outcome).

Our PEO(Population, Exposure and Outcome) question to guide the systematic review was formulated as follows: in adult patients (18 years or above) with T2DM, studies assessing the association between GV and cognitive impairment published in English or Chinese languages, acute GV evaluated with one or more parameters including MAGE, SD, CV or TIR using either self-monitoring of blood glucose (SMBG) or continuous glucose monitoring (CGM). 

2.3.2. In the "bias risk assessment" section, include the score range of the Newcastle-Ottawa Scale (0-9), to facilitate the reader's interpretation. Risk of bias assessment table can be added as supplementary material.

Response: In the "bias risk assessment" section, the score range of the Newcastle-Ottawa Scale (0-9) has been included, to help the readers to better understand.. We have added the risk of bias assessment table as supplementary material.

2.3.3. In the results synthesis section, reference values are indicated to interpret "r" as correlation, it is pertinent to place reference values as effect size, in which r= 0.1, r=0.3 and r=0.5 are considered small, moderate and large respectively.

Response: We modified the synthesis section, in which r= 0.1, r=0.3 and r=0.5 are considered correlations of small, medium and large respectively.

2.4. Results

2.4.1. Verify data. In the text it indicates that 64 studies were reviewed, but in figure 1 46 reviewed studies are mentioned.

Response: We have checked the data and had the errors modified.

2.4.2. Mention the relationship of the data found and the impact of the association, before the significance values, for example, "a lower cognitive performance was found in the subjects with greater glucose variation, which has statistical significance... and an effect size…”.

Response: We have added the interpretation of the relevant statistic values before the significance values.

2.4.3. Also mention the negative results of the glucose variability indicators in the text.

Response: The negative results of the glucose variability indicators have been added into the last paragraph of the discussion section.

2.4.4. Figure 2 can be omitted.

Response: Figure 2 has been deleted.

2.4.5. Add a subgroup analysis with the MMSE and MoCA.

Response: We have added a subgroup analysis with the MMSE and MoCA in the text.

2.5. Discussion

2.5.1. Write paragraphs of 4 to 8 lines to facilitate reading.

Response: For readers’ convenience, we have refined and reduced the content of the discussion to 4 to 8 lines.

2.5.2. It is necessary to write in the first paragraph the main results, that is, the relationship between glycemic variability and cognitive performance. Not only that 9 studies were found.

Response: We have added the relationship between glycemic variability and cognitive performance in the first paragraph of discussion.

2.5.3. It is not relevant to indicate the brand of the glucose monitor (“Abbott” flash glucose monitor) and it creates a conflict of interest, so I suggest removing it.

Response: We have removed the brand name of the glucose monitor.

2.5.4. Differences were found between the indicators of glucose variability and its relationship with cognitive functions. Although there are few studies, a description or explanation of why there are differences between the indicators can be made. 

Response: We have added descriptions and interpretations of the different indicators in the discussion section.

2.5.5. The paragraph on the controversies regarding the relationship of glucose variability to complications delves into cardiac problems that are not relevant to the topic, so it is necessary to explicitly state their relevance to cognitive performance or eliminate that information.

Response: We have removed the discussion of heart issues unrelated to the subject matter.

2.5.6. The writing can be finished highlighting the main results and adding the clinical implications.

Response: We have highlighted the major results and added the clinical significance in the end of text.

---

## [Decision Letter · Decision Letter 1]

26 Jul 2023

Relationship between Acute Glucose Variability and Cognitive decline in Type 2 Diabetes: A Systematic Review and Meta-Analysis

PONE-D-23-09111R1

Dear Dr. Haiyan Chi

We’re pleased to inform you that your manuscript has been judged scientifically suitable for publication and will be formally accepted for publication once it meets all outstanding technical requirements.

Kind regards,

Victor Manuel Mendoza-Nuñez, PhD

Academic Editor

PLOS ONE

---

## [Editor Report · Acceptance letter]

24 Aug 2023

PONE-D-23-09111R1 

Relationship between Acute Glucose Variability and Cognitive decline in Type 2 Diabetes: A Systematic Review and Meta-Analysis 

Dear Dr. Chi:

I'm pleased to inform you that your manuscript has been deemed suitable for publication in PLOS ONE. Congratulations! Your manuscript is now with our production department. 

Kind regards, 

on behalf of

Dr. Victor Manuel Mendoza-Nuñez 

Academic Editor

PLOS ONE